

# Detecting and quantifying methane emissions from oil and gas production: algorithm development with ground-truth calibration based on Sentinel-2 satellite imagery

Zhan Zhang[1], Evan D. Sherwin[1], Daniel J. Varon[2, 3], and Adam R. Brandt[1]

[1]Department of Energy Resources Engineering, Stanford University, Stanford, California 94305, United States
[2]School of Engineering and Applied Science, Harvard University, Cambridge, 02138, United States
[3]GHGSat, Inc., Montréal, H2W 1Y5, Canada

**Correspondence:** Adam R. Brandt (abrandt@stanford.edu)

**Abstract.** Sentinel-2 satellite imagery has been shown by studies to be capable of detecting and quantifying methane emissions from oil and gas production. However, current methods lack performance validation by calibration with ground-truth testing. This study developed a multi-band-multi-pass-multi-comparison methane retrieval algorithm that enhances Sentinel-2 sensitivity to methane plumes. The method was calibrated using data from a large-scale controlled release test in Ehrenberg,

Arizona in fall 2021, with three algorithm parameters tuned based on the true emission rates. Tuned parameters are the pixel-level concentration upper bound threshold during extreme value removal, the number of comparison dates, and the pixel-level methane concentration percentage threshold when determining the spatial extent of a plume. We found that a low value of the upper bound threshold during extreme value removal can result in false negatives. A high number of comparison dates helps enhance the algorithm sensitivity to the plumes in the target date, but values in excess of 12 days are neither necessary nor

computationally efficient. A high percentage threshold when determining the spatial extent of a plume helps enhance the quantification accuracy, but it may harm the yes/no detection accuracy. We found that there is a trade-off between quantification accuracy and detection accuracy. In a scenario with the highest quantification accuracy, we achieved the lowest quantification error and had zero false positive detections; however, the algorithm missed 3 true plumes which reduced the yes/no detection accuracy. On the contrary, all the true plumes were detected in the highest detection accuracy scenario, but the emission rate

quantification had higher errors. We also illustrated a two-step method that updates the emission rate estimates in an interim step which improves quantification accuracy while keeping high yes/no detection accuracy. We also validated the algorithm's ability to avoid false positives by applying it to a nearby region with no emissions.

## 1   Introduction

Methane ($CH_4$) emissions during oil and natural gas production are receiving increased attention since $CH_4$ is a potent green-

house gas (GHG) with radiative forcing 84 times greater than that of $CO_2$ over a 20-year time frame (MacKay et al., 2021). In recent years, fossil fuel (coal, oil and gas) production and use was estimated to have contributed 81-154 Tg $CH_4$ $a^{-1}$ of methane emissions, accounting for around one third of the global anthropogenic methane fluxes (Saunois et al., 2020). Another



estimate suggested that >80 Tg of methane emissions were from the oil and gas sector across the globe in 2021, $\sim 30\%$ higher than the 62 Tg in 2000 (IEA, 2022). The most detailed studies to date have been performed in the United States, where the methane loss rate from oil and gas supply in 2015 was estimated at 2.3% of the gross natural gas production (Alvarez et al., 2018). Studies also claim that the U.S. official inventories have been consistently underestimating methane emissions in oil and natural gas systems, suggesting a more important role for methane in GHG emissions reduction in the oil and gas sector (Alvarez et al., 2018; Brandt et al., 2014; Zavala-Araiza et al., 2015; Rutherford et al., 2021).

Reducing methane loss from oil and gas systems will require measurement and monitoring. Because of the large spatial scale of the oil and gas industry, there has been significant interest in methane measurement methods using aircraft or satellites to detect methane emissions across large areas (Karion et al., 2013; Hausmann et al., 2016; Frankenberg et al., 2016; Chen et al., 2022; Cusworth et al., 2021). Particularly, satellite detection has been considered a promising methane emissions monitoring technology because of its frequent revisit time, wide spatial coverage and low labor cost. SCIAMACHY (2003-2012) and Greenhouse Gases Observing Satellite (GOSAT, 2009-present) were the first two satellites to measure total methane columns by solar backscatter in the shortwave infrared (SWIR) (Jacob et al., 2016). The EO-1 Hyperion spectrometer achieved the first orbital detection of a methane superemitter plume from the Aliso Canyon release in 2016 (Thompson et al., 2016). The TROPOspheric Monitoring Instrument (TROPOMI) on the Sentinel-5 Precursor satellite launched in 2017 maps methane columns with daily global coverage at up to $7 \times 5.5 km^2$ resolution (Veefkind et al., 2012; Hu et al., 2018). The GHGSat constellation instruments launched from 2016-2022, each provide methane measurements with 25-50m spatial resolution over a $\sim 12 \times 12 km^2$ domain (Varon et al., 2018, 2020). More recently, the Sentinel-2 twin land-surveying satellites launched in 2015 and 2017 were shown to have moderate sensitivity to methane at specific wavelength bands (Varon et al., 2021). And other space-based sensors designed for land-surface monitoring, such as PRISMA (30m spatial resolution), Landsat-8 (30m spatial resolution), and WorldView-3 (WV-3, 3.7m spatial resolution), have similarly demonstrated methane detection capabilities (Cusworth et al., 2019; Ehret et al., 2021; Sánchez-García et al., 2022). Several studies in the last few years have reported methane enhancements from oil and gas producing regions and monitored methane "ultra-emitters" from oil and gas production based on the data from these satellite instruments (Lauvaux et al., 2022; Ehret et al., 2021; Irakulis-Loitxate et al., 2022; Cusworth et al., 2021).

The Sentinel-2 constellation has two polar-orbiting satellites placed in the same sun-synchronous orbit phased at $180°$ to each other. The main Sentinel-2 data products are imagery from 13 spectral bands from the visible to the SWIR (Phiri et al., 2020). Among these spectral bands, bands 11 ($\sim 1560-1660$ nm) and 12 ($\sim 2090-2290$ nm) integrate radiances over methane's 1650 and 2300 nm SWIR absorption features, thus enabling methane detection and quantification. Because of its global coverage, fine spatial resolution ($20 \times 20 m^2$ in band 11 and 12) and frequent revisit time (2-5 days), Sentinel-2 is believed to have potential for large-scale high-frequency monitoring of methane plumes in oil and gas producing regions (Ehret et al., 2021).

Varon et al. (2021) developed three retrieval approaches to derive methane enhancements across a scene of a methane point source based on the Sentinel-2 data in bands 11 and 12. The single-band-multi-pass (SBMP) retrieval method uses the changes in band-12 reflectance between a satellite pass with a plume and a pass sampling a reference scene with no plume to derive methane column enhancements. The multi-band-single-pass (MBSP) retrieval compares reflectance in band 11 and





12 on a single pass. And the multi-band-multi-pass (MBMP) retrieval applies two MBSP retrievals on two satellite passes to

remove artifacts from the retrieval field. In that work, two case studies of applying these approaches to methane point-source

plume detection from oil and gas facilities were presented, one in the Hassi Messaoud oil field of Algeria and the other in
the Korpezhe oil and gas field of Turkmenistan. The Korpezhe retrieval results were shown to be consistent with GHGSat-D
satellite instrument observations in 2018-2019 although with higher observation density. Among the three retrieval methods,
MBMP method generally performs the best, mainly because it increases the contrast of the plumes by combining two spectral
bands and having one pass sampling a reference scene.

However, the retrieval methods from Varon et al. (2021) might still be improved. First, calibration of the retrieved emission
source rates with ground-truth values needs to be done to validate the performance of the sensor and the retrieval method.
Varon et al. (2021) validated the retrieval results by comparing them with GHGSat observations since GHGSat has relatively
higher precision; however, ground-truth calibration with controlled release volumes is still essential in performance validation
and retrieval method fine tuning. Second, the retrieval methods include tunable parameters such as the percentage threshold

during plume mask extraction. Nevertheless, the optimal values of the tunable parameters were not discussed. Lastly, because
of Sentinel-2's limited sensitivity to methane, the MBMP retrieval method can generate false detections if the atmospheric
conditions between satellite passes are different or if some ground features have higher reflectance in band 11 than band 12.
And removing these false detections still relies on manual verification, such as checking if a similar shape occurs in the satellite
observation of the other bands or in the imagery basemap. New modifications need to be made to remove the false detections

at scale in a reasonable and convenient way.

Here we present a Multi-band-multi-pass-multi-comparison (MBMPMC) retrieval algorithm based on the MBMP approach
from Varon et al. (2021). The new algorithm extends the MBMP approach to enhance its sensitivity to methane plumes and
reduces false detections. Additionally, we were able to calibrate the method using data from a single-blind controlled release
test in Ehrenberg, Arizona in fall 2021. During calibration, three algorithm parameters were tuned based on the ground-

truth emission rates to improve the algorithm performance. Furthermore, we show two simple application studies of the new
algorithm, one in an extended time period at the same region with the controlled release test, and one at a different region in
the same time period. To our knowledge, this is the first time that a methane detection and quantification algorithm based on
Sentinel-2 imagery has been calibrated with ground-truth emission rates.

## 2  Methodology

### 2.1  MBMPMC retrieval algorithm

The MBMPMC retrieval algorithm is an improved retrieval method with modifications based on the MBMP retrieval method
from Varon et al. (2021). The new algorithm follows the same logic of retrieving the vertical column concentrations of at-
mospheric methane $\Delta\Omega$ ($kg \cdot m^{-2}$) from Sentinel-2 SWIR reflectances (see Figure 1). The main idea is retrieving methane
column concentrations from one spectral measurement featuring methane absorption and one not, such as two observations

from different passes with or without a methane plume, or two adjacent spectral bands with different methane absorption



properties. For a given scene, the method compares the Sentinel-2 measurements with the top-of-atmosphere (TOA) radiance simulated by a 100-layer, clear-sky radiative transfer model at 0.02 nm spectral resolution over the band 11 and 12 wavelength ranges. The specific steps are: first in a specific pass 1, the methane concentration enhancements are retrieved by minimizing the difference between the fractional change of Sentinel-2 reflectance and a fractional absorption model based on the simulated

TOA radiance in bands 11 and 12; then the same process is repeated in another pass 2, and the difference of these two retrieved column enhancements (two MBSP retrievals) is the MBMP methane column enhancement in pass 1 (Equation (1)). Here the subtraction between two passes aims to remove systematic errors in the MBSP retrieval due to wavelength separation between bands 11 and 12. In other words, the MBSP retrieval in pass 2 is mainly used for removing artifacts of the MBSP retrieval in pass 1. Therefore, in this paper we name pass 1 as the "target date (TD)" and pass 2 as the "comparison date (CD)" for

clarification. The TD in our method is the date for which the plume size is estimated. And by default here the target date is assumed to be chronologically after the comparison date, although in practice this need not be the case.

$$\Delta\Omega_{MBMP} = \Delta\Omega_{MBSP,TD} - \Delta\Omega_{MBSP,CD}(kg \cdot m^{-2}) \tag{1}$$

We make some modifications during the column retrieval process since the MBMP retrieval can still lead to false detections, especially in the MBMP subtraction step (Equation (1)). In theory, in the background with no methane plume, we expect the

two MBSP retrievals to have similar values of methane column enhancements since they are at the same scene. However, this is not always true because: (1) MBSP retrieval can be greatly affected by the atmospheric conditions such as cloud coverage; (2) the MBSP retrieval in one pass may have similar spatial distribution but with all the pixel values higher or lower than the MBSP retrieval in another pass due to differences in various atmospheric or earth properties (e.g., solar zenith angle, surface albedo) between different dates; and (3) other unpredictable random measurement errors can occur in a specific pass. Therefore, we

add the following steps to further reduce the number of false detections (see Figure 1 for sequence):

*Choose clear-view passes*. First, we only select passes with a clear view for both the target date and comparison dates since clouds can result in false detections by affecting reflectance. Here we use Sentinel-2 cloud probability, a data product created with the sentinel2-cloud-detector library, to select clear-view passes with no large cloud coverage. Specifically, we select the passes with less than 10% of cloud coverage (i.e., the area with cloud probability higher than 65% is less than 10% of the total

area of the study region).

*Normalization*. If two MBSP retrievals of Equation (1) have a uniform value difference in all the pixels, artifacts will still be preserved after the MBMP subtraction. So we normalize both MBSP retrievals before the MBMP subtraction to maximize the effects of artifacts removal. For example, in Figure 2, the MBMP retrievals with normalization show more plume contrast with the background compared with the ones without normalization. Some artifacts, such as the straight line in the unnormalized

retrieval with 09/19/2021 as the comparison date, are also removed in the normalized retrieval. Therefore, changing MBSP retrievals to the same scale helps enhance the ability to detect true methane plumes. However, note that the resulting concentration enhancements after normalization are no longer "actual" enhancements, thus should not be used to calculate the emission rates. In other words, normalization is only used for detecting the plume location and shape.



*Remove extreme values.* In some cases extremely high methane column enhancements can be generated for a small number
of pixels because of the appearance of random features in one of the two passes. Thus we also remove extreme values for the
two MBSP retrievals before normalization. The removal method is based on setting upper and lower bound thresholds, and
truncating values outside the bound thresholds to the threshold values. Here we set the lower bound threshold as $0 \, \text{kg m}^{-2}$, and
the upper bound threshold will be tuned using the controlled release experimental data below. Similar with normalization, this
step is only used for plume detection instead of quantification.

*Include multiple comparison dates.* Instead of using a single comparison date, we include multiple comparison dates to
help with plume detection. Different with the "sliding window" method from Ehret et al. (2021) which uses a multi-linear
regression onto 1-20 previous passes, we directly take the average of comparison date retrievals as the subtrahend in the
MBMP subtraction. Using multiple comparison days helps to stabilize the background since the background values can vary
among different passes due to weather, temperature, surface albedo difference, and other variation. Shown in Figure 3, more
comparison dates provide a more stable background, and therefore are more likely to increase the contrast of the plumes. On
the other hand, it is possible that in real application, the comparison date may also have methane plumes at the same location
with similar shape as the plumes in the target date. In this case, it is harder for the algorithm to detect the target date plumes
after the MBMP subtraction. So using the average of multiple comparison dates helps lower the possibility of the occurrence
of high-volume methane plume in the subtrahend, thus enhance the algorithm sensitivity to the plumes in the target date. Here
the comparison dates are selected as continuous clear-view passes before the target date, and the number of comparison dates
is a parameter that will be tuned using the controlled release experimental data below. Because the new algorithm considers
multiple comparison dates for the multi-band-multi-pass approach, it is named the "Multi-band-multi-pass-multi-comparison"
(MBMPMC) retrieval algorithm.

After column retrieval, the methane column enhancements $\Delta\Omega_{MBMPMC}$ are further used to calculate the emission source
rate $Q$ using the Integrated mass enhancement (IME) method described by Varon et al. (2021) (Equation (2)) (Frankenberg
et al., 2016; Varon et al., 2018). In this equation, IME is the integrated mass enhancement (kg), $U_{eff}$ is the effective wind
speed (m/s), and $L$ is the plume size (m).

$$Q = 3.6 \times \frac{IME \times U_{eff}}{L} (t/hr) \qquad (2)$$

To calculate IME, we first generate Boolean plume masks based on $\Delta\Omega_{MBMPMC}$ by selecting methane columns above
some percentage threshold for the scene, and smooth with a $3 \times 3$ median filter and a $3 \times 3$ Gaussian filter (see Figure1 (e)).
Here the percentage threshold is a parameter that will be tuned using the controlled release experimental data below. This
plume mask generation step sets the location and shape of the methane plumes.

Then the IME is defined as the sum of multiplication of column enhancements and pixel-level area of all the mask pixels.
Note that the column enhancements here are the original enhancements without any data transformation such as normalization
or extreme value removal applied to aid detection of the plume shape. The effective wind speed $U_{eff}$ is the function of
the local 10 m wind speed $U_{10}$ derived by Varon et al. (2021), calibrated with large-eddy simulations. We collect local wind



speed data from the High-Resolution Rapid Refresh (HRRR) atmospheric model from U.S. National Oceanic and Atmospheric Administration(U.S. NOAA, 2021). The plume size $L$ is taken in a simplified form as the square root of the plume mask area.

## 2.2 Performance assessment

To validate the performance of the new algorithm, calibration is required to compare the algorithm outcome with the ground truth. The goal of calibration is to assess the algorithm performance in both detection and quantification. Accurate yes/no detection is defined as the algorithm being able to detect a methane plume when it appears, and detecting nothing when no plume appears. Accurate quantification means that the emission rate estimates derived from the algorithm are consistent with the ground-truth measured release volumes.

Additionally, the algorithm performance can also be improved by parameter tuning to best match the ground truth. Here three parameters in the new algorithm are tuned: (1) the upper bound threshold during extreme value removal $b_u$, (2) the number of comparison dates for each target date $n$, and (3) the percentage threshold during the plume mask generation $p$. The way each parameter affects the algorithm outcome is described as below:

*The upper bound threshold $b_u$*: $b_u$ is a parameter that occurs during the extreme value removal, during which the retrieval
values higher than it are considered to be extreme outliers and are replaced by the threshold value. So a lower $b_u$ means a more strict constraint during extreme value removal. Ideally, an optimal $b_u$ helps remove false detections due to the extreme highs. However, if $b_u$ is too low, a true methane plume may also be ignored since its retrieval values could be removed.

*The number of comparison dates $n$*: We expect that the higher $n$ is, the more stable the background is, thus the contrast of the plume is increased. However, this stability increase is not linear, so the increase of $n$ may not help much in the case of a
very large $n$. In addition, the computation workload also increases along with higher $n$, approximately linearly with $n$.

*The percentage threshold $p$*: The higher $p$ is, the fewer pixels are included in the plume mask. So a higher $p$ means a smaller plume mask area. This may help with removing false positives and enhancing quantification accuracy, but may also lead to false negatives or result in underestimation of plume volume if selected at too high of a value.

To quantify the algorithm performance, we use two assessment factors with focus on different aspects. First, we choose F1
score to assess the performance of detection. F1 score is a function of "precision" and "recall", measures of false positives and false negatives respectively (Equations (3)(4)(5)). F1 score has a range of 0 to 1, with higher values representing better algorithm performance. In addition, we choose the average absolute error (AAE) to assess the performance of quantification (Equation (6), where $x_i$ and $\hat{x_i}$ are the emission rate estimate and ground-truth emission rate in day $i$, and $N$ is the number of days). AAE has a range of 0 to $\infty$ with lower values suggesting better algorithm performance. Absolute error is used so that
under- and over-estimates do not cancel each other out.

$$F1 = 2 \times \frac{\text{precision} \cdot \text{recall}}{\text{precision} + \text{recall}} \qquad (3)$$

$$\text{precision} = \frac{\#\text{True Positive}}{\#\text{True Positive} + \#\text{False Positive}} \qquad (4)$$



$$recall = \frac{\#\text{True Positive}}{\#\text{True Positive} + \#\text{False Negative}} \tag{5}$$

$$AAE = \frac{\sum_{i=1}^{N} |x_i - \hat{x}_i|}{N} \tag{6}$$

## 3 Results

In fall 2021, a single-blind controlled release test was conducted by the Stanford University Environmental Assessment & Optimization Group. The test was performed in Ehrenberg, Arizona, the testing methods are described in detail in Rutherford et al. (2022), and the test was generally similar to previous tests of airplane-based methane plume detection from the same group (Sherwin et al., 2021). This test aimed at assessing the performance of various aircraft and satellite methane detection technologies. During the test, the participants were given the information of time and location of the potential release, although the methane plume volumes (including zero, i.e., no methane plume) were unknown to them. Participants were asked to estimate the mass emissions rate during each observation in kg $CH_4$/h. Specifically for Sentinel-2, there are 7 clear-view satellite passes and one cloud-covered pass covered in this test from 10/17/2021 to 11/03/2021. Here we consider only the 7 clear-view passes, and also add three dates after the test with zero emission, so that in total 10 target dates with ground-truth emission rates are used to do the ground-truth calibration. Of the 10 target dates, 5 have methane plumes with non-zero emission rates, and 5 have no methane plumes. Figure 4 region A is the study region that covers the controlled release point source. After calibration, we also provided two simple application studies to validate the algorithm performance (Section 3.2). Because we lacked other ground-truth data to use as a blind test set, one goal of these application studies was to test if the algorithm can avoid generating false positives in the case of no methane plumes.

### 3.1 Controlled release calibration

We selected a wide value range for each algorithm parameter during the parameter tuning. For $b_u$, we noticed that the magnitudes of the pixel-level column enhancements of a methane plume are usually from $10^{-3}$ to $10^{-1} kg \cdot m^{-2}$. So we selected 10 values from 0.01 to 0.1 $kg \cdot m^{-2}$ with increment 0.01 $kg \cdot m^{-2}$, and 4 other values 0.005, 0.12, 0.15 and 0.20 $kg \cdot m^{-2}$. For $n$, for each target date 15 clear-view passes were selected with the earliest comparison date around 45 days before the target date, so n ranges from 1 to 15 with increment 1. And for $p$, 16 values were selected from 0.80 to 0.95 with increment 0.01. Therefore, there are in total 3360 scenarios of different combinations of three parameters. Each of these 3360 parameter settings was run to quantify volumes from all 10 study days.

Figure 5 shows how each parameter affects the algorithm outcome. In each figure, an assessment factor (AAE or F1 score) is shown as a function of two parameters, based on a fixed value of the third parameter (i.e., a "slice" through 2 parameters keeping the third constant). Here the fixed values are from the parameter setting with the lowest AAE. Figures 5(a)(b) show that a small $b_u$ value (0.005-0.02 $kg \cdot m^{-2}$) leads to bad algorithm performance with high AAE and low F1 score (AAE>1.3,





F1 score<0.4). This suggests that the $b_u$ constraint is too strict in this range and removes retrievals not only from the extreme highs, but also from true methane plumes. Thus the algorithm starts to generate false negatives. Particularly in Figure 5(b) when $b_u$ is $0.005 kg \cdot m^{-2}$, we see NAN values of F1 score because there is no true positive detection at all. Aside from the low value range, AAE and F1 score show less sensitivity to $b_u$ at the other values. Therefore, the conclusion from $b_u$ tuning is that one should avoid excessively low values of $b_u$ ($< 0.02 kg \cdot m^{-2}$).

Figures 5(a)(c) show a rough decreasing trend of AAE along with higher $n$ when $n < 12$. This suggests that a higher $n$ helps with quantification accuracy by providing a more stable background and lowering the possibility of high-volume plume in the comparison dates. However, AAE does not show an obvious decrease when $n \geq 12$, which suggests that 12 or more comparison dates are not necessary, or at least ceases to improve performance. Figure 5(b)(d) show low F1 scores when $n$ is low (for example, F1 scores <0.67 when $n = 2$). This is because some target dates have their earlier comparison dates with higher methane plume volumes, and a low value of $n$ does not effectively reduce the average volume in the comparison dates, thus resulting in more false negatives. In real application, this may be a more serious problem if the plume is continuous among a long time period with varying volumes. Additionally, computational cost is roughly proportional to $n$, so too high of a value of $n$ can have excessive computational costs with little benefit to accuracy. Therefore, the value of $n$ should not be too low nor too high, and from the figures we can conclude that a reasonable choice of $n$ is in the range 10-12.

Figure 5(c)(e) show that AAE decreases with higher $p$ at first, but starts to increase when $p > 0.92$. The decreasing trend is due to smaller plume volumes and less false positives resulting from smaller plume masks during the Boolean plume mask generation. The increasing trend in high $p$ range, however, is because $p$ becomes sufficiently high such that no mask is generated even for the dates with real methane plumes. This also explains why in Figure 5(d)(f) the F1 score is low in high $p$ ranges. Low AAEs occur in the $p$ range 0.91-0.93, while high F1 scores occur in the $p$ range 0.85-0.86. This suggests a trade-off between accurate quantification and accurate yes/no detection: accurate quantification usually requires a high $p$ value, but accurate yes/no detection needs a lower $p$ value (though not excessively low). Therefore, when selecting the best $p$ value, we can choose to emphasize quantification accuracy and accept the possibility of missing plumes ($p > 0.90$) ; or we can choose to detect more plumes, and accept the possibility of emission rate overestimation($p \approx 0.85$).

Here two specific scenarios shown in Table 1 and 2 further illustrate the trade-off between accurate quantification and accurate yes/no detection. The "Min AAE" scenario is an example of pursuing quantification accuracy. It has the lowest AAE of all the parameter settings and with the highest precision, meaning that it also has the minimum amount of false positives. However, this scenario has three false negatives that reduce the F1 score. Aside from this specific scenario, the top 1% scenarios with low AAEs have their $b_u$ ranging widely in $0.03 - 0.15$, $n$ in a middle-to-high range of $7 - 14$ and $p$ staying high in $0.91 - 0.92$. On the other hand, the "Max F1 score" scenario has the highest F1 score. It doesn't have false negatives, but in order to find all plumes it becomes too aggressive, leading to one false positive. Note that multiple scenarios have the same highest F1 score, and the scenario we show here is the one with the lowest AAE among them. The top 1% scenarios with high F1 scores have their $b_u$ ranging widely in $0.02 - 0.12$, $n$ in a wide range of $1 - 15$ and $p$ in the middle range of $0.82 - 0.85$.

As a compromise, we developed a method to apply approaches in sequence to reduce the quantification error further while keeping a high F1 score. The specific steps are: (1) apply a scenario with high F1 score as the base case to generate the first




round of emission rate estimates; (2) raise the value of $p$ and apply the updated scenario again to generate the second round of emission rate estimates; (3) for the passes with non-zero emission rates in both scenarios, update the base case estimates to the new ones since they are likely to be closer to the ground-truth volumes. We name this method the "two-step application"

method. Here we only change the value of $p$ since the mask extraction step where $p$ is applied is after the column retrieval step where $b_u$ and $n$ are applied. So a consistent $b_u$ and $n$ greatly reduces the computation workload as we only need to redo the mask extraction. Table 1 shows an example of the two-step application ("Two-step hybrid" scenario) with the "Base case" scenario. Results show that the 'Two-step hybrid' scenario achieves lower AAE than the "Base case" scenario with F1 score remaining the same. Specific locations and shapes of detected plumes in "Min AAE", "Max F1 score" and "Two-step hybrid"

scenarios are shown in Figure 6.

We also compared the performance of MBMPMC algorithm with MBMP, MBSP and SBMP methods from Varon et al. (2021) in Figure 7. The top row is for a true emission rate of 7.38 tCH$_4$/h while the bottom row is for a true emission rate of 0 tCH$_4$/h. Results show that the MBMPMC algorithm performs the best with both true positive and true negative detections. Its emission rate estimates are also the closest to the ground-truth volumes. The MBMP method has true negative detection in

10/17/2021, but shows a small false positive detection in 10/19/2021. Its emission rate estimate in this date is also much lower than the ground truth. MBSP and SBMP retrievals perform worst with multiple large-area false positive plumes.

### 3.2    Broader application to examine false positives in cases of no ground release

To test the algorithm's performance in avoiding false positives, we applied the algorithm with the "Min AAE" scenario since it achieved zero false positive in the ground-truth calibration above. Two application studies were designed, one in an extended

three-month time period from 10/01/2021 to 12/31/2021 at the same region with the controlled release test (Figure 4 region A), and one at a different region (Figure 4 region B) in the same time period. The algorithm shows zero emission in all the passes of both two studies, which validates its performance of avoiding false positives. Two detection examples are shown in Figure 8.

### 4    Conclusion

This study presented a multi-band-multi-pass-multi-comparison (MBMPMC) methane retrieval algorithm using Sentinel-2 satellite imagery with several modifications based on the multi-band-multi-pass (MBMP) retrieval method from Varon et al. (2021). The major modification is including multiple comparison dates into the retrieval, which helps increase the contrast of the plume by stabilizing the background.

The new retrieval algorithm was then calibrated by a controlled release test in Ehrenberg, Arizona in fall 2021. During cali-

bration, three algorithm parameters were tuned based on the ground-truth emission rates to improve the algorithm performance. They are the the pixel-level concentration upper bound threshold $b_u$ for extreme value removal, the number of comparison dates $n$, and the pixel-level methane concentration percentage threshold $p$ when determining the spatial extent of a plume. We found that although the algorithm sensitivity to $b_u$ is generally not very high, a low $b_u$ value can decrease its accuracy by resulting in





false negatives. $n$ value should be high enough to enhance the algorithm sensitivity to the plumes in the target date, but values
$> 12$ are neither necessary nor computationally efficient. A high $p$ value helps enhance the quantification accuracy, but it may
harm the yes/no detection accuracy by missing some true plumes.

The controlled release calibration suggests that there is a trade-off between quantification accuracy and detection accuracy. If
the algorithm aims to guarantee the quantification accuracy and avoid false positives, then a $b_u$ in range 0.03-0.15, a $n$ in range
7-14 and a $p$ in range 0.91-0.92 are preferable. If the algorithm is expected to guarantee the detection accuracy, particularly
with the fewest false negatives, then it would be more appropriate to choose $b_u$ in 0.02-0.12, $n$ in range 1-15 and $p$ in range
0.82-0.85. We also illustrate a two-step method that changes the parameter values and updates the emission rate estimates in
an interim step which improves quantification accuracy while keeping high yes/no detection accuracy.

To our knowledge, this is the first study that validates the performance of a Sentinel-2 methane detection and quantification
algorithm by calibrating it with the ground-truth emission rates. We believe the ground truth calibration offers researchers an
opportunity to optimally tune methane retrieval algorithms and have confidence in their widespread deployment. In the future,
the MBMPMC algorithm can be validated with more systematic experiments wherein the algorithm can be adjusted or tuned
to meet different detection expectations.

We believe that the algorithm can still be improved further in the following aspects. First, the optimal values of three
parameters may vary in different situations. For example, $b_u$ may vary with the methane plume volumes; $n$ is affected by
whether the plume is continuous or discrete in time; and $p$ also depends on the area of the plume and the area of the study
region, so it may vary with the study region size. This study simplifies these problems since our controlled release test covers
only one region with a short time period. We hope to explore these questions more with more abundant ground-truth data in
the future. Additionally, the current algorithm focuses more on removing false positives resulting from the background noise
of the comparison dates. In real applications, however, more false positives due to the background noise of the target dates may
be generated. Removing these false positives requires more work after the plume mask generation, such as removing the plume
masks that are far away from known well pad or pipeline locations. Other options may involve applying machine vision based
shape learning methods to filter out plume masks with shapes unlikely to be generated by a gas cloud. We hope to develop an
efficient method of false detection removal so that Sentinel-2 can play a more important role in routine oil and gas methane
monitoring in the global scale.

*Code and data availability.* The methane detection and quantification algorithm code will be made available upon request. The methane
column retrieval code will be made available for non-commercial use upon request (GHGSAT Data and Products – Copyright © 2021
GHGSAT Inc. All rights reserved). The Sentinel-2 satellite imagery are available in the Google Earth Engine (GEE) cloud platform, and the
HRRR wind data are available in the AWS HRRR GRIB2 Archive. Both the data collection codes will be made available upon request.



*Author contributions.* ZZ, EDS, DJV and ARB contributed to the study conceptualization. ZZ conducted controlled release calibration and
application studies, and wrote the manuscript with review and edits from all the other authors.

*Competing interests.* There are not conflicts of interest to declare.

*Acknowledgements.* This work is funded by California Air Resources Board (Agreement No. 18ISD011). The authors also acknowledge
ExxonMobil and the Stanford Strategic Energy Alliance for funding the Ehrenberg controlled release test



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



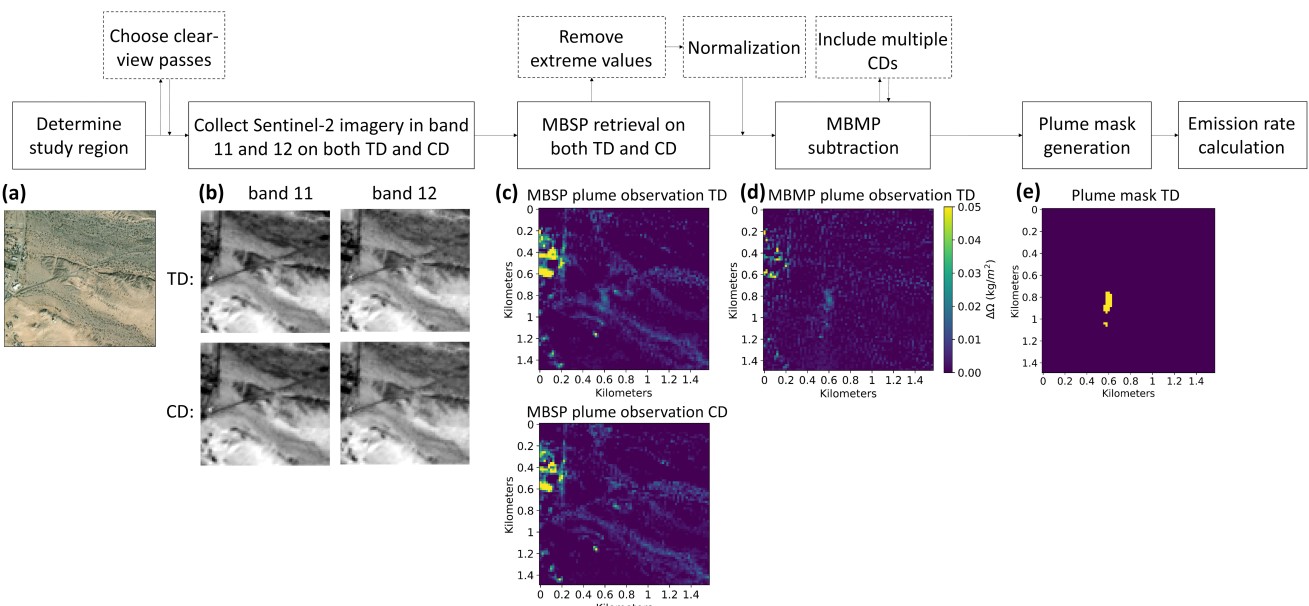

**Figure 1.** Basic algorithm workflow. Solid boxes are specific steps of the multi-band-multi-pass (MBMP) retrieval. Dashed boxes are new modifications added in this study. (a): study region. (b): Sentinel-2 imagery in band 11 and 12 on both target date (TD, top row) and comparison date (CD, bottom row), with pixel value as reflectance. (c): MBSP retrieval on both TD (top row) and CD (bottom row) with pixel value as methane column concentration ($kg \cdot m^{-2}$). (d): MBMP retrieval on TD, i.e., the result of subtracting the MBSP retrieval on TD by the MBSP retrieval on CD. (e): Boolean plume mask generated from MBMP retrieval by selecting methane columns above some percentage threshold for the scene, and smooth with a median filter (window size $3 \times 3$) and a Gaussian filter (window size $3 \times 3$). Basemap of (a): ArcGIS Online World Imagery Basemap. Sources: Esri, DigitalGlobe, GeoEye, i-cubed, USDA FSA, USGS, AEX, Getmapping, Aerogrid, IGN, IGP, swisstopo, and the GIS User Community.





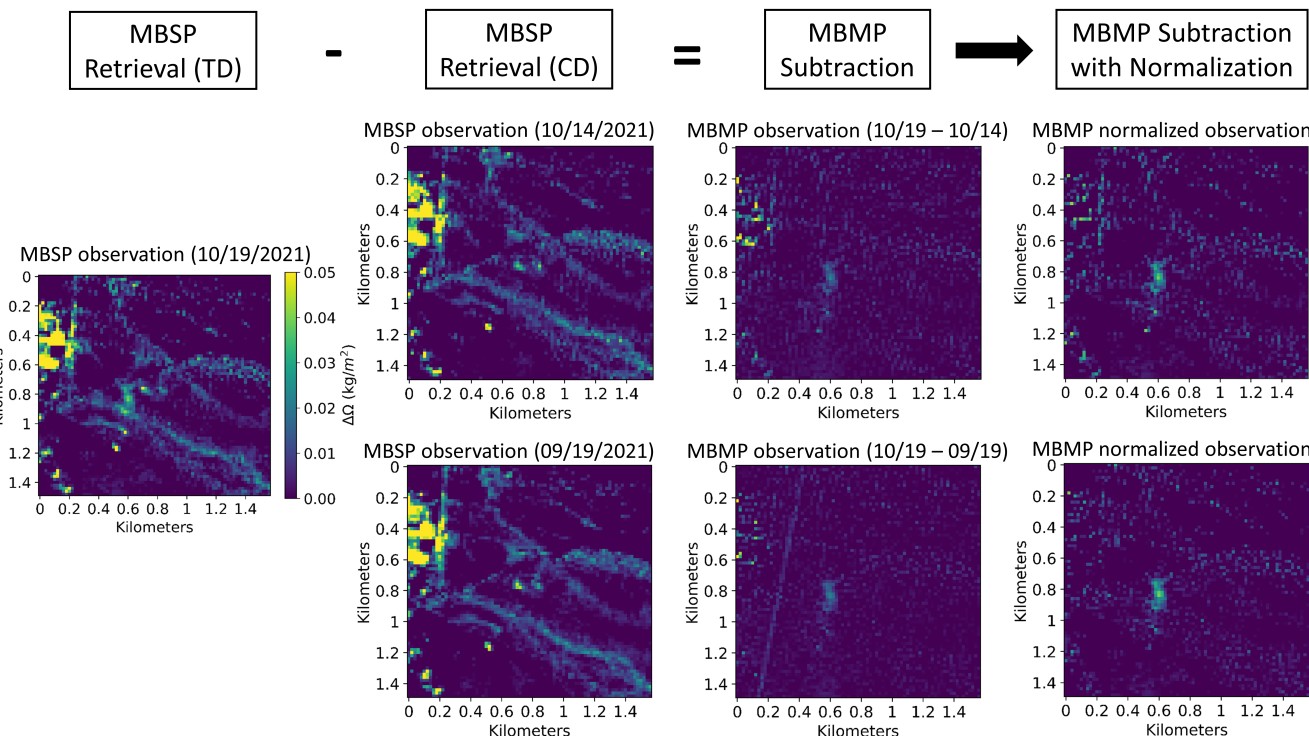

**Figure 2.** Examples of normalization. Here are two MBMP retrieval examples, one with 10/19/2021 as target date (TD) and 10/14/2021 as comparison date (CD), and the other with 10/19/2021 as TD and 09/10/2021 as CD. In each example, we show MBMP plume observation without and with normalization (see figures of the third and fourth column from left). In both examples, the normalized MBMP retrieval shows more plume contrast with the background than the one without normalization. Particularly in the second example with 09/10/2021 as CD, there is a straight line artifact in the MBMP retrieval without normalization, and it is removed in the normalized MBMP retrieval. This illustrates the fact that normalization improves the effect of artifacts removal by making MBSP retrievals of TD and CD to the same scale.



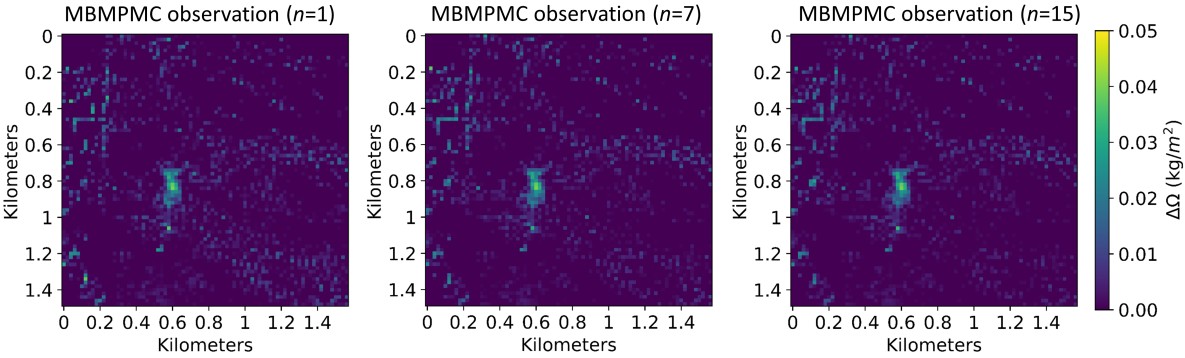

**Figure 3.** Examples of including multiple comparison dates. In the MBMP subtraction, we include multiple comparison dates and take their average MBSP retrievals as the subtrahend to stabilize the varying background in different dates. Here $n$ is the number of comparison dates. From left to right figures ($n = 1$, 7 and 15) we can see that a higher $n$ provides a "cleaner" background in the MBMPMC retrieval, particularly in the lower right area, thus increases the contrast of the plume.





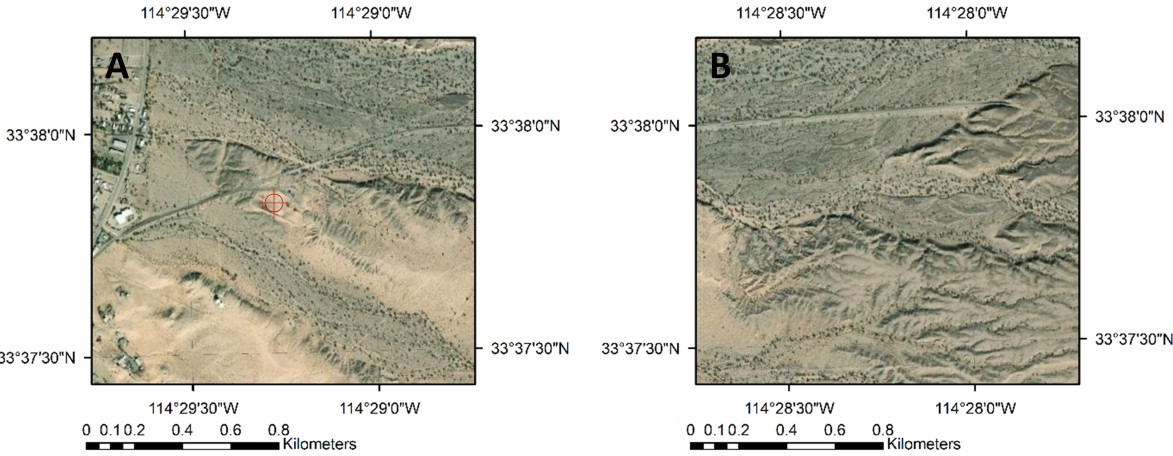

**Figure 4.** Study regions. Region A covers the controlled release point source (red-marked, 33.6306°N, 114.4878°W) and is mainly used for the controlled release calibration. Region B is to the east of region A with the same area, and is used for the application study. Basemap: ArcGIS Online World Imagery Basemap. Sources: Esri, DigitalGlobe, GeoEye, i-cubed, USDA FSA, USGS, AEX, Getmapping, Aerogrid, IGN, IGP, swisstopo, and the GIS User Community.





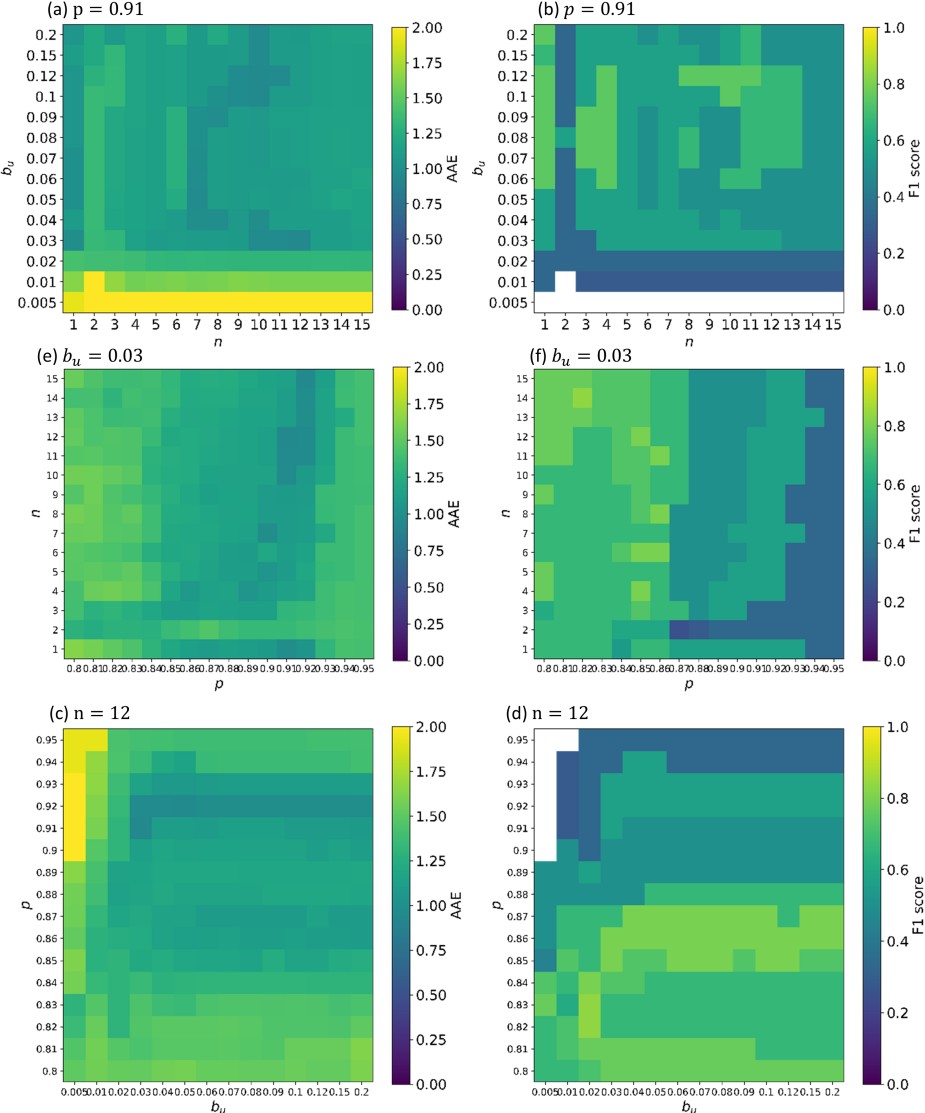

**Figure 5.** Parameter tuning scenarios. Three algorithm parameters are tuned: the upper bound threshold during extreme value removal $b_u$, the number of comparison dates for each target date $n$, and the percentage threshold during the plume mask generation $p$. Two assessment factors are used with focus on different aspects: the average absolute error (AAE) assesses quantification performance, and F1 score assesses detection performance. Totally 3360 parameter settings were run with wide value ranges of three parameters. Figures (a)(b) show how AAE and F1 score change with $b_u$ and $n$ respectively with $p = 0.91$; (c)(d) show how two assessment factors change with $n$ and $p$ with $b_u = 0.03$; (e)(f) show how they change with $p$ and $b_u$ with $n = 12$. The fixed values are from the parameter setting with the lowest AAE. White space indicates NAN value of F1 score resulted from zero true positive.




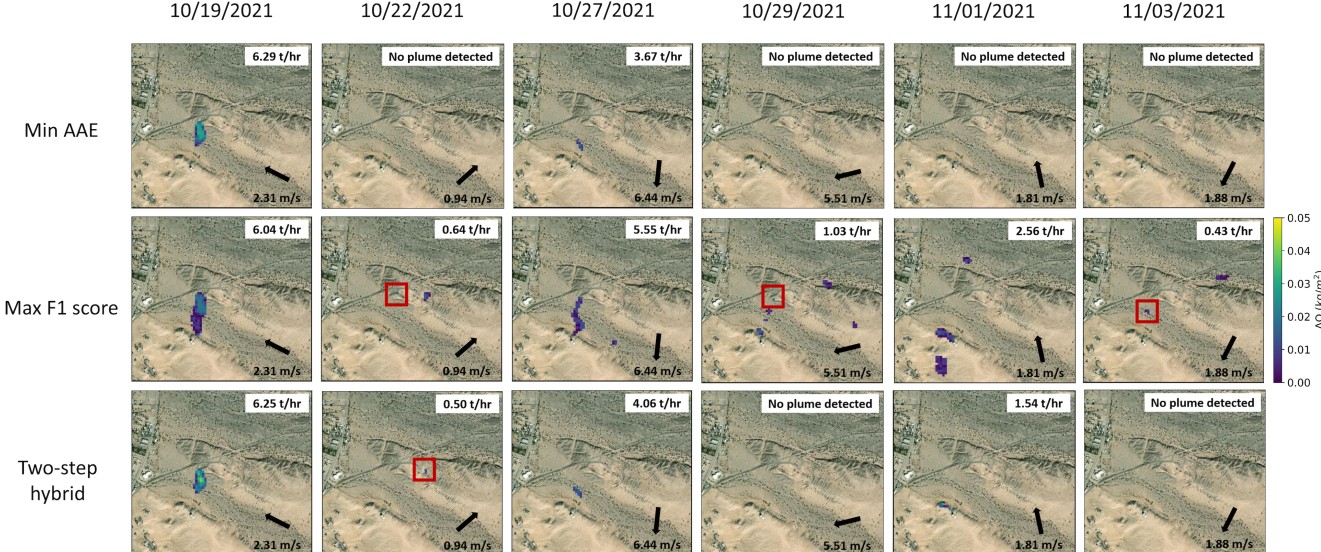

**Figure 6.** Locations and shapes of the detected plumes. All the dates with plumes detected in "Min AAE", "Max F1 score" and "Two-step hybrid" scenarios are shown. Plumes that are too small with only few pixels are red-marked, although they are not necessarily the full plume extent. Each figure also has methane plume emission rate shown in the upper right area, and wind speed and direction shown in the lower right. Note that the plumes in 11/01/2021 are false positives since the ground-truth volume in this day is 0, and the dates with multiple plumes detected are also likely to include false positives although it's hard to validate. Basemap: ArcGIS Online World Imagery Basemap. Sources: Esri, DigitalGlobe, GeoEye, i-cubed, USDA FSA, USGS, AEX, Getmapping, Aerogrid, IGN, IGP, swisstopo, and the GIS User Community.





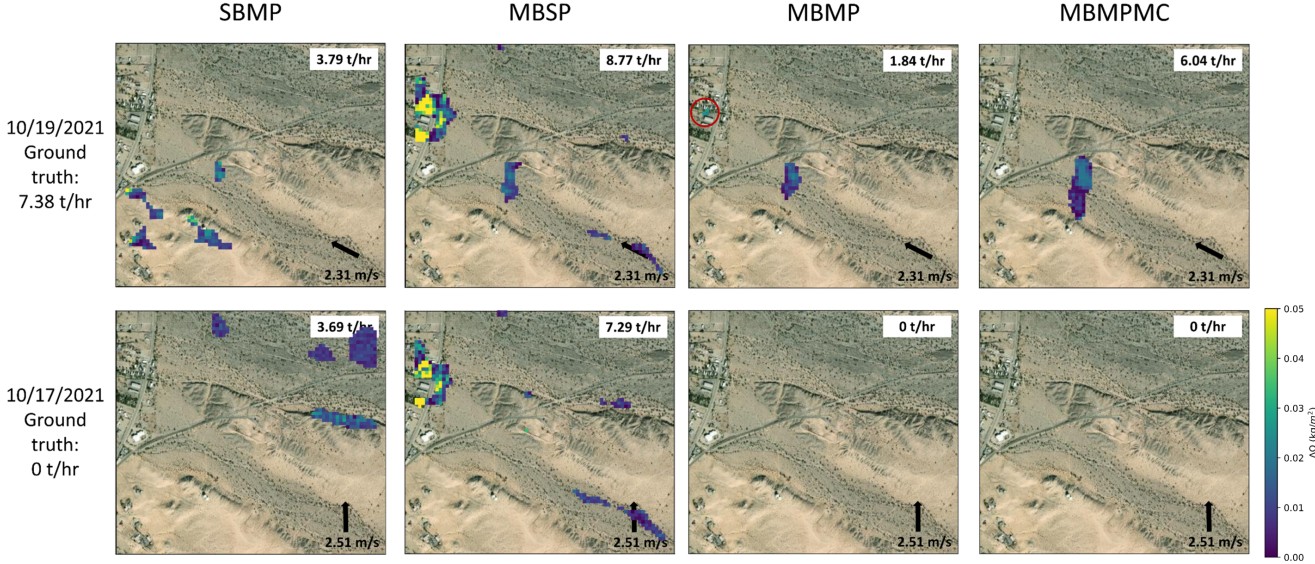

**Figure 7.** Comparison of four retrieval methods. The performance of multi-band-multi-pass-multi-comparison (MBMPMC) algorithm is compared with multi-band-multi-pass (MBMP) method, multi-band-single-pass (MBSP) method and single-band-multi-pass (SBMP) method from Varon et al. (2021) in two dates, one with methane plume (10/19/2021) and one with no plume (10/17/2021). The MBMPMC algorithm performs the best with correct yes/no detection and emission rate estimates closest to the ground-truth volumes. The MBMP retrieval has a small-area false positive detection in 10/19/2021 (red-circled), and its emission rate estimate is much lower to the ground truth. MBSP and SBMP methods perform the worst with multiple large-area false positive plumes. Basemap: ArcGIS Online World Imagery Basemap. Sources: Esri, DigitalGlobe, GeoEye, i-cubed, USDA FSA, USGS, AEX, Getmapping, Aerogrid, IGN, IGP, swisstopo, and the GIS User Community.



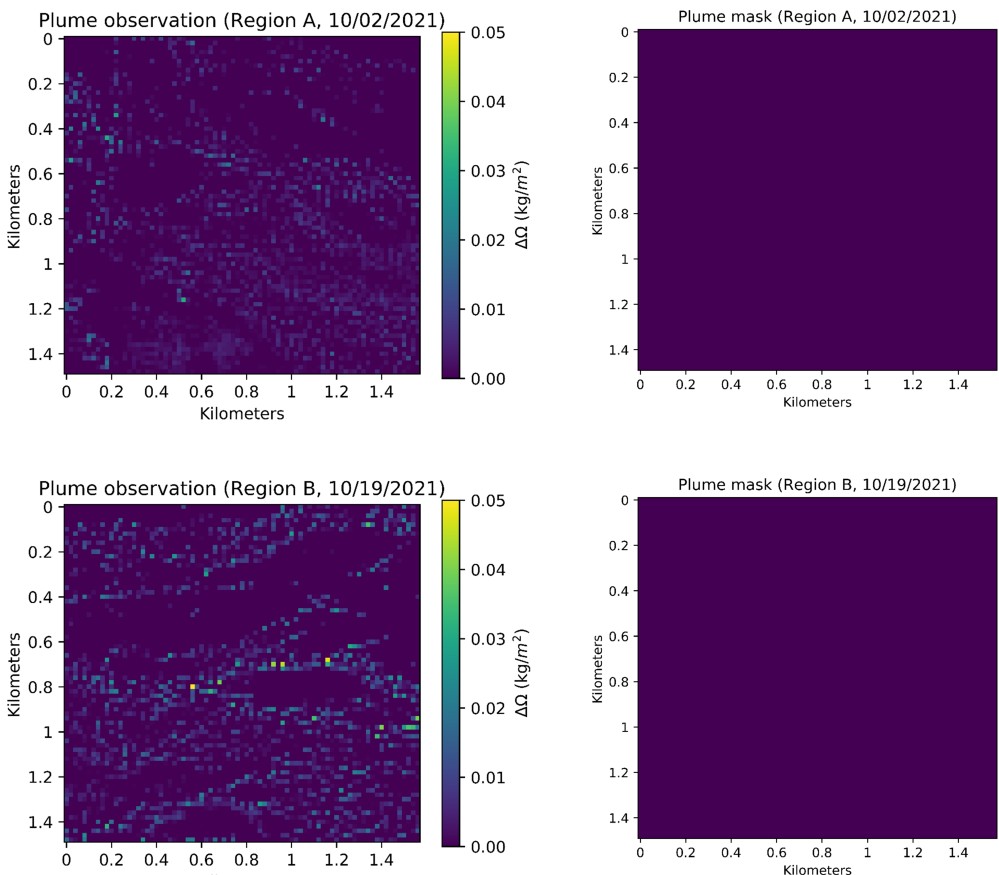

**Figure 8.** Two examples of application studies. Top row shows an application case at 10/02/2021 in region A, and bottom row shows a case at 10/19/2021 in region B. Both examples have true negative detection. All the other dates in the application studies have true negative detection as well.



**Table 1.** Scenario examples of three parameters.

| Scenarios | $b_u$ | $n$ | $p$ | AAE | F1 score | #False positives | #False negatives |
|---|---|---|---|---|---|---|---|
| Min AAE | 0.03 | 12 | 0.91 | 0.94 | 0.57 | 0 | 3 |
| Max F1 score | 0.02 | 14 | 0.84 | 1.20 | 0.91 | 1 | 0 |
| Base case | 0.04 | 10 | 0.87 | 1.18 | 0.67 | 1 | 2 |
| Two-step hybrid | 0.04 | 10 | $0.87 \rightarrow 0.91$ | 1.09 | 0.67 | 1 | 2 |

"Min AAE" scenario: the scenario with the lowest AAE; "Max F1 score" scenario: the scenario with the highest F1 score; "Base case" scenario: the base case of the two-step application method example; "Two-step hybrid" scenario: the two-step application method example.



**Table 2.** Emission rate estimates of scenario examples.

|  | Ground truth (t/hr) | Min AAE (t/hr) | Max F1 score (t/hr) | Base case (t/hr) | Two-step hybrid (t/hr) |
|---|---|---|---|---|---|
| 10/17/2021 | 0 | 0 | 0 | 0 | 0 |
| 10/19/2021 | 7.38 | 6.29 | 6.04 | 6.07 | 6.25 |
| 10/22/2021 | 1.69 | 0 | 0.64 | 0.50 | 0.50 |
| 10/27/2021 | 3.60 | 3.67 | 5.55 | 4.86 | 4.06 |
| 10/29/2021 | 5.18 | 0 | 1.03 | 0 | 0 |
| 11/01/2021 | 0 | 0 | 2.56 | 1.54 | 1.54 |
| 11/03/2021 | 1.40 | 0 | 0.43 | 0 | 0 |
| 11/06/2021 | 0 | 0 | 0 | 0 | 0 |
| 11/08/2021 | 0 | 0 | 0 | 0 | 0 |
| 11/11/2021 | 0 | 0 | 0 | 0 | 0 |

"Min AAE" scenario: the scenario with the lowest AAE; "Max F1 score" scenario: the scenario with the highest F1 score; "Base case" scenario: the base case of the two-step application method example; "Two-step hybrid" scenario: the two-step application method example.