# Peer review of "Detecting and quantifying methane emissions from oil and gas production: algorithm development with ground-truth calibration based on Sentinel-2 satellite imagery"

_EGUsphere, 2022_

## Author Comment (AC1)

**Responses to reviewers: "Detecting and quantifying methane emissions from oil and gas production: algorithm development with ground-truth calibration based on Sentinel-2 satellite imagery"**

We appreciate the reviewers for their comments and revision suggestions. Below we offer our point-by-point responses explaining how we addressed the comments.

The revised text as it appears in the manuscript is written in normal blue text, with new contents underlined and removed contents struck through.

**Response to comments from Anonymous Referee #1**

The authors developed a multi-band-multi-pass-multi-comparison methane retrieval algorithm that enhances Sentinel-2 sensitivity to methane plumes. The new algorithm is based on the algorithm developed by the same author but enhances its sensitivity to methane plumes and reduces false detections. The manuscript is well written. The method looks sound. I recommend publication after minor revision.

General comments:

1. Section 3.1. Line 250-259. What is the relationship between "two-step application" and MBMPMC? It is not clear to me.

   *Answer:* The "two-step application" method is a way we propose to apply the MBMPMC algorithm in order to achieve higher quantification accuracy while keeping high yes/no detection accuracy. In the "two-step application" method, we run the MBMPMC algorithm to get the first round of emission rate estimates, and then redo the plume mask extraction step by raising the value of parameter $p$, and finally update the emission rate estimates. Different with direct application of the MBMPMC algorithm, this method is specifically designed to address the trade-off issue between quantification accuracy and detection accuracy. We updated the text in L250-257 to make it clearer for the readers.

   As a compromise, we developed a method to apply  the MBMPMC algorithm in sequence to reduce the quantification error further while keeping a high F1 score…So a consistent bu and n greatly reduces the computation workload as we only need to redo the mask extraction. Different with direct application of the MBMPMC algorithm, this method is specifically designed to address the trade-off issue between quantification accuracy and detection accuracy. Table 1 shows an example of the two-step application ("Two-step hybrid" scenario) with the "Base case" scenario.

2. Figure 7. The improvement compared to the previous 3 methods are very impressive. Please try to briefly summarize the reasons for the improvement compared to each method.

   *Answer:* Thanks for the suggestion. We added a summary of the improvement reasons into the manuscript (L. 264-266).

The MBMP method has true negative detection in 10/17/2021, but shows a small false positive detection in 10/19/2021. Its emission rate estimate in this date is also much lower than the ground truth. This implies that the steps of normalization and inclusion of multiple comparison dates in the MBMPMC method contribute to a higher sensitivity to the true plume than the MBMP method. MBSP and SBMP retrievals perform worst with multiple large-area false positive plumes. SBMP method is likely to produce false detections if the surface albedo changes across different passes, and MBMPMC method reduces the effect of changing surface albedo by including different spectral bands and multiple comparison dates. MBSP method can produce false detections because of the wavelength separation between two spectral bands, and MBMPMC method largely removes these artifacts by subtracting the MBSP retrieval between different passes.

Specific comments:

1. There are two "also" in the last two sentences. Please try to rephrase them.

   *Answer:* We rephrased the last two sentences in Abstract (L. 15-18).

   We  illustrated a two-step method that updates the emission rate estimates in an interim step which improves quantification accuracy while keeping high yes/no detection accuracy. We also validated the algorithm's ability to avoid false positives by applying it to a nearby region with no emissions.

2. It is useful to clarify the ratio of anthropogenic to natural emissions as well.

   *Answer:* Thanks for the suggestion. We added a sentence clarifying the anthropogenic methane emissions ratio.

   During the 2008-2017 decade, around 60% of global methane emissions are from anthropogenic sources (Saunois et al., 2020). Of these sources,  fossil fuel (coal, oil and gas) production and use was estimated to have contributed 81-154 Tg $CH_4$ $a^{-1}$ of methane emissions, accounting for around one third of the global anthropogenic methane fluxes (Saunois et al., 2020).

3. Section 2.1. Please mention that there is a flow chart to illustrate the steps of MBMPMC when first discussing them.

   *Answer:* We added a sentence mentioning the flow chart in the manuscript.

   The new algorithm follows the same logic of retrieving the vertical column concentrations of atmospheric methane $\Delta\Omega$ (kg · $m^{-2}$) from Sentinel-2 SWIR reflectances . Main steps are shown in the flow chart of Figure 1.

---

## Author Comment (AC3)

**Responses to reviewers: "Detecting and quantifying methane emissions from oil and gas production: algorithm development with ground-truth calibration based on Sentinel-2 satellite imagery"**

We appreciate the reviewers for their comments and revision suggestions. Below we offer our point-by-point responses explaining how we addressed the comments.

The revised text as it appears in the manuscript is written in normal blue text, with new contents underlined and removed contents struck through.

**Response to comments from Anonymous Referee #2**

"The manuscript by Zhang et al. deals with methane plume retrievals with the Sentinel-2 satellite mission. They use methane concentration enhancement maps derived from Sentinel-2 data over controlled methane releases to constrain free parameters in the retrieval and emission rate estimation algorithms. They show that the Sentinel-2 detection and quantification of methane plumes from those controlled releases improves after model calibration with the same in situ data.

In my opinion, the research discussed in this manuscript must be of interest to the methane remote sensing community, especially considering the recent and rapid development of satellite-based high-resolution methane mapping methods. Also, the topic fits perfectly in AMTD, where the first paper on the use of Sentinel-2 for methane mapping (Varon et al., 2021) was published.

On the other hand, I have some major concerns with the overall purpose and some technical details of this work. In particular, I am not sure about the value of calibrating the algorithms with ground truth in this case. Is there any hope that they can be extrapolated to other sites or even seasons at the same site? I would say no. The algorithm parameters that they are optimizing are strongly acquisition dependent. For example, the thresholds accounting for outliers and false positives are driven by surface characteristics (homogeneity, stability). The finding that 12 dates are optimal for the multitemporal method wouldn't apply to a site with changing vegetation covers, for which a configuration with one recent reference acquisition would be better than with a combination of 12 of them. Also, the thresholds used to filter out outliers should depend on the heterogeneity of the site. The retrieval noise, and hence the spatial extent of the plume, will also depend on the surface heterogeneity.

This extrapolation question would also apply to the proposed two-step method to improve emission quantification. Is this relevant to a wide community if data from a controlled methane release are needed as input?"

*Answer:* Thank you for raising this point. As you note, this study is based on a controlled release test with stable surface characteristics, which is more favorable for methane remote sensing than a heterogeneous sites with changing vegetation covers or a new/disappearing structure (smoke, flaring, etc.). How to filter out outliers from site heterogeneity is one of the key questions we are interested in, which we can only partially answer for now due to the site limitation of our first-phase controlled release test. However, given that we are currently conducting two months of

follow-up controlled release tests, which will be followed in coming years with additional sites in different regions and with different surface features, we believe that there will be more room to explore ways of background noise removal in various heterogeneous sites. Based on these tests, we plan to further update the methodology and application of this study with consideration of more complex background features in future papers. That said, we believe the results we present in this paper stand on their own as a significant scientific contribution.

In this study, we added more discussion of the limitations of this study and potential improvement in the future (see below).

We believe that the algorithm can still be improved further in the following aspects. First, the optimal values of three parameters may vary in different situations. For example, $b_u$ may vary with the methane plume volumes; $n$ is affected by whether the plume is continuous or discrete in time; and $p$ also depends on the area of the plume and the area of the study region, so it may vary with the study region size. In particular, this study is based on a homogeneous study area and results may not generalize to heterogeneous sites with changing surface features during the study time period (e.g. due to seasonal shifts in vegetation). How to filter out outliers and define the true plume in a heterogeneous site is still difficult to answer since our controlled release test covers only one region over a single month. In future controlled release tests, we hope to explore these questions further based on more abundant ground-truth data in areas with more complex background features. Additionally, the current algorithm focuses more on removing false positives resulting from background noise in images from different comparison dates. In real applications, however, another concern will be false positives due to the background noise in images collected on the target dates. Removing these false positives requires more work after the plume mask generation, such as removing the plume masks that are far away from known well pad or pipeline locations. Other options may involve developing an automatic approach of outlier filtering and plume definition, as in Ehret et al. (2021), or applying machine vision based shape learning methods to filter out plume masks with shapes unlikely to be generated by a gas cloud. We hope to develop an efficient method of false detection removal so that Sentinel-2 can play a larger role in routine oil and gas methane monitoring across the globe.

"I think that the authors should show that the estimated model parameters can be applied outside this particular experiment for this work to be relevant. I don't think that the no false-negative test in Fig. 8 is a proper assessment of the model extrapolation that I am asking for (no emission to evaluate, and acquisition conditions for site B are very close to those of site A).

Perhaps the authors could run tests of how those thresholds perform for other sites, especially those with a more complex surface such as the US sites included in Ehret et al. https://pubs.acs.org/doi/10.1021/acs.est.1c08575?"

*Answer:* Thanks for the suggestion. Following your suggestion, we run tests on a methane-emitting site in the Permian basin during July-September 2020 studied in Ehret et al. Using the parameters of the "Max F1 score" scenario, which showed the highest detection accuracy in the controlled release test, we detected all the 9 plumes represented in Ehret et al. The detected plumes have similar shapes as those detected in the original paper, although the emission rate estimates have a discrepancy. We believe that this new true-positive test, along with the no falsenegative test in the manuscript, provide evidence in support of the detection accuracy of our method. We expect the quantification accuracy to be analyzed in more detail in future, larger-scale controlled release tests.

**3.2 Broader application  in cases of unknown emission rates**

**3.2.1 Examine true positives**

To test the algorithm's performance in detecting true positives, we applied the algorithm in a methane-emitting site in the Permian basin during the summer of 2020 studied in Ehret et al. (2021). We used the parameters of the "Max F1 score" scenario which achieved the highest detection accuracy in the ground-truth calibration above. We detected all plumes from the 9 days covered in Ehret et al. (2021) with similar plume shapes and the emission rate estimate difference within ±55%. This test validates the performance of detecting true positives of our method (Figure 8).

[Figure]

**Figure 8.** Examine true positives. The application site is in the Permian basin during the summer of 2020 studied in Ehret et al. (2021) (31.7335∘N, 102.0421∘W). The first and third rows show plume observation of this study, and the second and fourth row show plume observation from Ehret et al. (2021) (image source: Ehret et al. (2021)). All the 9 plumes represented in Ehret et al. (2021) were detected with similar shapes in this study. The emission rate estimate difference is within ±55% of Ehret et al. (2021).

"In addition, I think it should it be possible to use this nice dataset to investigate possible approaches for automatic estimation of the outlier filtering threshold and the plume definition threshold. Those thresholds should be based on scene-based noise/heterogeneity estimates, such as n-sigmas above the retrieval noise level. Perhaps the authors could come up with approaches to estimate those parameters from the methane enhancement maps and cross-compare the results with the values derived from the model calibration presented in the manuscript. Examples of such threshold estimation approaches can be found in Ehret et al. (Background Estimation)."

*Answer:* Thanks for the suggestion. It is inspiring to see both the two-step estimation method (including a step of discarding the most extreme 5% pixels) and the clustering method in Ehret et al. having impressive effects in outlier filtering and plume definition. This implies that an automatic approach of outlier filtering and plume definition is possible, and this is particularly important for a heterogeneous site with changing surface features. As mentioned above, we plan to update this methodology with consideration of more complex surface features based on more ground truth data from future controlled release tests, and it is definitely worth exploring ways to do automatic outlier filtering and plume definition in the future method updating, although doing so would be outside the scope of this paper. We now briefly raise this idea in the manuscript, which is shown in our first answer (also as below).

Other options may involve developing an automatic approach of outlier filtering and plume definition as in Ehret et al. (2021), or applying machine vision based shape learning methods to filter out plume masks with shapes unlikely to be generated by a gas cloud.

Other comments:

1. Sec 3.1, List of steps to improve plume detection: most of those steps (clear-view overpasses, normalization, removal of outliers, multiple reference data) are relatively obvious and already included in existing algorithms (e.g. Ehret et al., Gorrono et al.). Because of this, I am not sure that the methodology presented in this manuscript deserves to be presented as a new method, including an own acronym.

   *Answer:* Thanks for the comment. We would like to use the new acronym to differentiate this method from the earlier MBSP, SBMP and MBMP methods in Varon et al. as this method is an update of the earlier ones. Since the acronym "MBMPMC" may be too long, we decided to change it to "MBPD" (as "multi-band, multi-pass, multi-dates") to make it more reader friendly.

2. L2. (and L159) "performance validation by calibration" – not sure what this means

   *Answer:* Thanks for the comment, this sentenced was rephrased.

   However, current methods lack performance  calibration with ground-truth testing.

3. L288 – Not sure about statements on thresholds and detection accuracy: comments might apply to the avoidance of false negatives, but not the occurrence of false positives (which is actually the main difficulty for plume detection in real detection scenarios).

   *Answer:* Thanks for the comment, indeed this sentence may cause confusion to readers, so we rephrased it as below.

   If the algorithm aims to guarantee the quantification accuracy , then a $b_u$ in range 0.03-0.15, a $n$ in range 7-14 and a $p$ in range 0.91-0.92 are preferable.

4. Units: a space is missing between numbers and units, such as in "30m"

   *Answer:* Thanks for the comment, the units were revised with space added.

5. L145. Please, check citations.

   Shouldn't this preprint on the Stanford methane release experiment be cited https://eartharxiv.org/repository/view/3465/?

   *Answer:* Thanks for the comment, the citation has been corrected.

   The test was performed in Ehrenberg, Arizona, the testing methods are described in detail in Sherwin et al. (2021) and Rutherford et al. (2022).